# A Calibration Procedure Which Accounts for Non-linearity in Single-monochromator Brewer Ozone Spectrophotometer Measurements

*Zahra Vaziri Zanjani* [1], *Omid Moeini* [1], *Tom McElroy* [1], *David Barton* [1], *Vladimir Savastiouk* [2]

[1] York University, 4700 Keele Street, Toronto, Ontario, M3J 1P3, Canada

[2] Full Spectrum Science Inc., 112 Tiago Avenue, Toronto, Ontario, Canada

*Correspondence to:* Zahra Vaziri Zanjani (zahra_vaziri@yahoo.com)

**Abstract.** It is now known that Single-monochromator Brewer Spectrophotometer ozone and sulphur dioxide measurements suffer from non-linearity at large ozone slant column amounts due to the presence of instrumental stray light caused by scattering within the optics of the instrument. Because of the large gradient in the ozone absorption spectrum in the near-ultraviolet, the atmospheric spectra measured by the instrument possess a very large gradient in intensity in the 300 to 325 nm wavelength region. This results in a

significant sensitivity to stray light when there is more than 1000 Dobson Units (DU) of ozone in the light path. As the light path (airmass) through ozone increases, the stray light effect on the measurements also increases. The measurements can be on the order of 10% low for an ozone column of 600 DU and an airmass factor of 3 (1800 DU slant column amount) which is an example of conditions that produce large slant column amounts.

Primary calibrations for the Brewer instrument are carried out at Mauna Loa Observatory in Hawaii and Izana Observatory in Tenerife. They are done using the Langley plot method to extrapolate a set of measurements made under a constant ozone vertical column to an extraterrestrial calibration constant. Since the effects of a small non-linearity at moderate ozone paths may still be important, a better calibration procedure should account for the non-linearity of the instrument response. Studies involving the scanning of a laser source have

been used to characterize the stray light response of the Brewer (Fioletov *et al.*, 2000), but until recently these data have not been used to elucidate the relationship between the stray light response and the ozone measurement non-linearity.

In a study done by Karppinen et. al. (2015), a method for correcting stray light has been presented that uses an additive correction which is determined via instrument slit characterization and a radiative transfer model

simulation and is then applied to the single Brewer data (Karppinen et al., 2015). The European Brewer Network is also applying stray light corrections which includes an iterative process that results in correcting the single Brewer data to agree with double Brewer data (Rimmer et al. 2018; Redondas et al. 2018). The first model requires measurements of the slit function and the latter method relies on a calibrated instrument, such as a double Brewer, to characterize the instrument and to determine a correction for stray light.

This paper presents a simple and practical method of correcting for the effects of stray light that includes a mathematical model of the instrument response and a non-linear retrieval approach that calculates the best values for the model parameters. The model can then be used in reverse to provide more accurate ozone values up to a defined maximum ozone slant path. The parameterization used was validated using an instrument physical model simulation. This model can be applied independently to any Brewer instrument and correct for the effects of stray light.

## 1. Introduction

The Brewer spectrophotometer instrument is a diffraction grating, polychromatic spectrophotometer which produces monochromatic light at a set of 6 exit slits. It measures total column amounts of ozone, sulphur dioxide and aerosol optical thickness (Silva and Kirchhoff, 2004) by direct sun measurements at five wave bands centered at the approximate wavelengths of 306.3, 310.1, 313.5, 316.8 and 320.0 nm (Kerr et al., 1984). The Brewer spectrophotometer was designed as a replacement for the Dobson instrument in the 1970s and became commercially available in the 1980s (Redondas et al., 2014).

There are two types of Brewers in use today: Single Brewers (Mk II, Mk IV, Mk V) and Double Brewers (Mk III). The double Brewer (DBr) spectrophotometer is the combination of two single Brewer (SBr) optical frames where the exit slits of the first monochromator are the entrance slits of the second one. The first monochromator disperses the light while the second one recombines it, giving the instrument a much greater capability for stray light rejection (Gröbner et al., 1998).

As in all optical instruments, stray light is also problematic in Brewer spectrophotometers. The source of stray light in a monochromator is light scattering from the various optical surfaces and walls of the instrument. Light scattered from the atmosphere is also a source of stray light which is called the sky-scattered radiation. Sky-scattered radiation can be eliminated by taking direct sun measurements very close to the sun and subtracting that from the solar measurements (Josefsson, 1992). Instrumental stray light becomes much more significant when the source of energy or detector sensitivity changes rapidly as a function of wavelength. In the Brewer, the source of energy is the sun or moon. When measuring ozone, as the solar zenith angle increases the importance of instrumental stray light increases, due to the increasing gradient in intensity towards longer wavelengths. One way to quantify the effect of stray light is to observe how the measurements deviate from linearity according to Beer's Law at large ozone slant paths (Slavin, 1963). In a Brewer spectrophotometer, instrumental stray light arises mainly from the holographic diffraction grating and the collimating and focusing mirrors (Silva and Kirchhoff, 2004). Comparisons between measurements of SBr and DBr have been made by Bais et al. (1996) which show a much lower effect of stray light in measurements from the DBr. At wavelengths below 300 nm the SBr shows a 10% underestimation in absolute irradiances when there is more than 1000 DU of ozone slant column present (Bais et. al., 1996). At wavelengths above 300 nm, the presence of stray light may be problematic as well. This paper addresses this issue.

According to the WOUDC website, over 200 Brewers, single and double, are being used in more than 40 countries and 100 stations to measure ozone column amounts (Savastiouk, 2006; WOUDC, 2016). Many single Brewers are being replaced by double Brewers, and therefore, new measurements may show a false increase in ozone column amounts due to the effect of lower stray light in the DBr, particularly for measurements made at large solar zenith angles. In this paper a method is described that accounts for the effect of stray light in the data from the SBr to provide more accurate, reprocessed SBr data.

## 2. Methodology and Data

This paper presents a practical method to improve Brewer measurements by compensating for the effect of stray light. This model expands the Langley method to include a term representing the contribution of the non-linearity in retrieved ozone resulting from the effects of stray light. The simulation of the Brewer performance using a physical model (Moeini et al., 2018) provided validation of the parameterization of the analysis model.

### 2.1. Physical Model

In the physical model, the slit function of the instrument measured is used to characterize the stray light. For this purpose, a HeCd laser at 325 nm was used as a source and all wavelengths starting at 290 nm for the SBr and DBr were measured. The photon count rate measured by the Brewer is the integral of the spectral intensities on all wavelengths weighted by the slit function. Figure 1 shows the measured slit function of a SBr #009 and DBr #119 reported by Moeini et. al. (2018). The dots show the measurements made by the HeCd laser and the solid lines show the best fit to the measurements. Under ideal conditions without the effect of stray light, the fit to the slit function would be a trapezoid with its wings extending to zero. But, in reality, as seen in the figure, the fit is trapezoidal from FWHM to the peak, but the wings form a Lorentzian function that extends to a horizontal line which is not at zero. The difference between the wings of the SBr and DBr slit functions clearly shows the presence of stray light. The horizontal fit line is approximately $10^{-4}$ for the SBr and $10^{-6}$ for the DBr which shows that the effect of stray light is much more pronounced in the SBr measurements than the DBr (Moeini et al., 2018).

### 2.2. Mathematical Model

To compensate for the effects of stray light on the measured ozone column amount a new technique is described in this paper. A sensitive method for measuring the effect of instrumental stray light is by using the deviation of the measurements from linearity according to Beer's law, at large ozone slant paths (Slavin, 1963).

Beer's law states that the attenuation of light by a material increases exponentially with an increase in path length in a uniformly absorbing medium. Equation (2.1) shows this relation where $I_0$ is the intensity of light before entering the layer of material, $I$ is the intensity of light after going through the layer of material in question, with an absorption coefficient of $\alpha$, a column amount of $X$ and a slant path length or airmass of $\mu$. In the Brewer instrument, $I_0$ is the intensity of the extra-terrestrial light before entering the atmosphere. This

absolute intensity measured by the ground-based instrument relies on the knowledge of the extraterrestrial source of the light, which is difficult to determine due to scattering and absorption by clouds and aerosols. Therefore, the absorbances at two different wavelengths, one short and one long, are calculated to construct a measurement function which is independent of absolute intensity. This method is called differential absorption

5      spectroscopy which is used for measurements made by the Dobson spectrophotometer. The Beer's law, which is linear with respect to airmass, can be calculated as shown in equation (2.2) where, $\Delta\alpha = \alpha_S - \alpha_L$, is the difference in absorption coefficients at $I_S$ and $I_L$, the intensities of the short and long wavelengths, and $I_{S0}$ and $I_{L0}$, the intensities of the short and long wavelength at zero airmass (no medium to absorb). The logarithm of the ratio of the two intensities is denoted as F and is called the absorption function shown in equation (2.3).

$$I = I_0 e^{-\alpha \cdot X \cdot \mu} \tag{2.1}$$

$$log\left(\frac{I_S}{I_L}\right) = -\Delta\alpha \cdot X \cdot \mu + log(\frac{I_{S0}}{I_{L0}}) \tag{2.2}$$

$$F = -\Delta\alpha \cdot X \cdot \mu + F_0 \tag{2.3}$$

In the Brewer instrument, five wavelengths are used instead of just the two. These wavelengths are 306.3, 310.1, 313.5, 316.8, and 320.1 nm. A weighting is applied to the logarithm of the counts at each wavelength and a sum is used to calculate the absorption function. This method eliminates the absolute dependence on

intensity and suppresses anything that is linear with respect to wavelength. This weighting will also produce measurements that are more sensitive to ozone. The weighting is shown in equation (2.4) where the absorption function, $F$, is calculated by the product of the weighting vector, $w$, by $L$ which is a 5 x 1 matrix composed of the logarithm of the counts measured at each wavelength, $log(C_\lambda)$, multiplied by $10^4$. A study done by Savastiouk and McElroy (2005) shows the calculations for deriving the weighting vector (Savastiouk and

McElroy, 2005)

$$F = w \cdot L = [0.0, \quad -1.0, \quad 0.5, \quad 2.2, \quad -1.7] \cdot log(C_\lambda) \cdot 10^4 \tag{2.4}$$

Under realistic conditions, Equation (2.3) deviates from its linear form at large airmass values in the presence of instrumental stray light. To account for this non-linearity, a new model for the absorption function measured by the instrument is defined. The form of the model, incorporating a correction to the absorption function which

is approximately cubic in ozone, was determined empirically by testing different corrections.

This new model accounts for the non-linearity in measuring ozone column amount. It also accounts for filter changes in the instrument. This instrument model is presented in equation (2.5) where, $F_m$ is the model absorption function, $\alpha$ is the absorption coefficient of ozone, $\mu$ is ozone airmass, $X$ is ozone column amount, $F_0$ is the absorption function at zero airmass, $\gamma$ is the non-linearity factor, $b_j$ is the filter change factor and $ND_j$

is the filter vector which is zero for filter numbers (j) not used and has a value of 1 for filter numbers used.

Different forms of the model have been experimented with and the model described in equation (2.5) was found to give good agreement with the observations.

$$F_m = F_0 - \alpha \cdot \mu \cdot x - \gamma \cdot (\alpha \cdot \mu \cdot x)^3 + \sum_j b_j \cdot ND_j \tag{2.5}$$

The components of the model (Eq. 2.5) to be determined are $v_k = (x, \gamma, b_j, F_0)$, where $k$ indicates the number of components of $v$ that are to be retrieved. The Langley method is used to determine the $k$ components of vector $v$ by finding suitable values for $v$ that minimize the square error between the model and the observations. This method is described in the following. In the first step, initial values for $v_k$ are estimated. To predict an initial value for $F_0$ the conventional Langley plot is used, where the absorption function versus airmass is plotted. As an example, the absorption function versus airmass of the single Brewer #009 is plotted in Figure 2, where the dots show the instrument measurements. The measured values tend to deviate from the linear model as the airmass increases. The plot is quite linear at airmass values smaller than 2, therefore, this part of the data is used to apply the least-squares method using the linear model (Eq. 2.3) to find the slope, $\alpha.x$ and the intercept at zero airmass, $F_0$. The initial value of $\gamma$ and $b_j$ should be very small therefore an initial value close to zero is used.

The non-linear Langley method uses the least-squares method to determine $v_k$. In this process the square error, $SE$, between the absorption function measured by the instrument, $F_i$, and the modelled absorption function, $F_{mi}$, for all the $N$ observations will be minimized. The index $i$ denotes the observation number from 1 to $N$. Therefore, the derivative of the difference of the observations and the modeled absorption function (Eq. 2.4) with respect to all the components of vector $v_k$ should be zero as shown in equation (2.6).

Because the model is non-linear in ozone, several iterations of the solution are made to arrive at an accurate result. The maximum number of iterations can be set at 50 or more but the answer reaches a useful convergence between 5 and 10 iterations. Equations (2.6) to (2.13) show the Langley method loop and the estimation of $v_k$. Where, $F_i - F_{mi}$ is replaced by $\Delta F_i$ in equation (2.7), $M_{ik}$ is the derivative of the model absorption function (Eq. 2.4) with respect to the k components of $v_k$ at each measurement point, $i$. $M_{in}$ denotes the derivative of the model with respect to the $n^{th}$ component of $v_k$ at each measurement point, $i$. The components of the jacobian, $M_{ik}$, have been provided in the appendix.

$$\frac{\partial}{\partial v_k}(SE) = \frac{\partial}{\partial v_k}\left[\sum_{i=1}^{N}(F_i - F_{mi})^2\right] = 0 \tag{2.6}$$

$$\frac{\partial}{\partial v_k}(SE) = 2 \cdot \sum_{i=1}^{N}\left[\Delta F_i - \frac{\partial(F_{mi})}{\partial v_n} \cdot \Delta v_n\right] \cdot \frac{\partial(F_{mi})}{\partial v_k} = 0 \tag{2.7}$$

$$M_{ik} = \frac{\partial(F_{mi})}{\partial v_k} \tag{2.8}$$

$$M_{in} = \frac{\partial(F_{mi})}{\partial v_n} \tag{2.9}$$

$$H = \sum_{i=1}^{N}[\Delta F_i \cdot M_{ik}] \tag{2.10}$$

$$K = \sum_{i=1}^{N}[M_{in} \cdot M_{ik}] \tag{2.11}$$

$$\Delta v_n = K^{-1} \cdot H \tag{2.12}$$

$$v_{k_{n+1}} = v_{k_n} + \Delta v_n \tag{2.13}$$

To improve accuracy, a weighting of the observations is needed. From Figure 2, it can be seen, that the measurements are denser at smaller airmass than at larger airmass therefore the model will put more emphasis on these points and estimate the components of $v_k$ accordingly. To reduce this problem the data are multiplied by $\frac{1}{\mu}$ and the uncertainty, $\frac{1}{\sigma^2}$, associated with the counts as shown in equation (2.14) before applying the model.

j is the number of wavelengths, which in this case is 1 to 5, Counts$_j$ is the photon count for each of the 5 wavelengths and $w_j$ is the weighting for each wavelength. With this weighting the difference between the model and measurements is exaggerated at large airmass and, therefore, carry more weight.

$$\sigma_i = \frac{1}{\sqrt{N_i}} = \frac{1}{\sqrt{\sum_{j=1}^{5}(Counts_{i,j} \cdot w_j)}} \tag{2.14}$$

### 2.3. Dead-time Correction

The dead-time of an instrument is important to account for. In a Brewer instrument the detector photomultiplier pulses have a finite width of ~30 ns. If two or more photons arrive at the detector within 30 ns they will be counted as one count. This will cause an error in counting the photons. This error associated with the dead-time increases with count rate (Kerr, 2010). The deadtime varies from instrument to instrument and can also change with time. For low ozone slant columns and high intensity solar radiation, a small change of 10 ns in the

deadtime can cause an error of up to 5% in the measured total ozone column amount (Fountoulakis et al., 2016). The dead-time correction is modelled in equation (2.15) where N is the measured photon count, $N_0$ is the corrected photon count and $\tau$ is the dead-time (Kerr, 2010).

$$N = N_0 \cdot e^{N_0 \cdot \tau} \tag{2.15}$$

To calculate dead-time, measurements are done with the Brewer through two exit slits simultaneously and separately through each exit slit. For example, measurements are done with exit slit 2 and 4 resulting in $N_2$ and

$N_4$ and measurements are done with both exit slits open resulting in $N_{2+4}$. If equation (2.15) is written for each case, 3 equations with 4 unknowns which are $N_{02}$, $N_{04}$, $N_{02+4}$ and $\tau$ result. A forth equation, $N_{02+4} = N_{02} + N_{04}$, will help find the dead-time (Kerr, 2010). This procedure is not done on a regular basis and the dead-time of an instrument changes somewhat over time, therefore a correction prediction for dead-time must be performed

during the Langley process. To account for the dead-time prediction, when calculating the derivative of the square error, *SE*, the derivative of $F_i$ with respect to dead-time must also be considered. In this case, $v_k$ is $\tau$ and thus Equation (2.8) becomes Equation (2.16) which shows how this derivative is included in the Langley method.

$$\boldsymbol{M_{ik}} = -\frac{\partial(\boldsymbol{F_i})}{\partial \boldsymbol{\tau}} \qquad\qquad\qquad (2.16)$$

**3.  Results and discussion**

The Langley method was applied to the data collected by two Brewer instruments #009 (Single Brewer) and #119 (Double brewer) stationed at Mauna Loa at -155.5° longitude and 19.5° latitude and as an example the results for 14th July 2010 and 25th October 2010 is presented. The non-linear mathematical model (Eq. 2.5) was applied to the data and model parameters where retrieved. The non-linearity term, filter change term and dead

time term were applied in reverse to the data to calculate corrected values for the absorption function and ozone values in Dobson units.

The observed, modelled and corrected absorption function (F) for Single Brewer #009 is plotted in Figure 3. In the observed and modelled F, you can clearly see the filter change causing a jump in the intensity of light received by the detector. This sudden fluctuation in intensity is modelled using the non-linear model and is

corrected to give a more linear F with respect to Ozone airmass. The same has been plotted for the Double Brewer #119 in Figure 4 where the observed modelled and corrected observations are very close to linear with respect to airmass and thus the correction is less than 1%.

Using the corrected absorption function the corrected ozone values where calculated. The measured and corrected ozone column amount in Dobson units versus ozone airmass for both the single (#009) and double

(#119) Brewers for 14th July 2010 and 25th October 2010 are illustrated in Figures 5 and 7 respectively. There is some nonlinearity observed for the single Brewer at low airmass which is corrected to an amount slightly higher than the double Brewer corrected amounts. The ratios of single to double Brewer measurements before and after applying corrections for 14th July 2010 and 25th October 2010 are plotted in Figures 6 and 8 respectively. The corrected ratio in red is closer to linearity than the uncorrected ratio in black. As a comparison

the obtained parameters from the model and the standard parameters used to retrieve ozone for 14th July 2010, can be seen in Table 1.

## 4. Conclusions

There is a difference of up to 10% in the ozone column amount measured by the single and double Brewers (single lower than double) as observed at large ozone slant paths and illustrated in Figure 5 for 14[th] July 2010 and Figure 7 for 25[th] October 2010. The mathematical model presented in this paper accounts for this non-linearity in the data collected by single Brewers. Applying corrections to the historical data of single Brewers is essential, as it will eliminate errors due to, the effect of stray light in the ozone measurements.

The double Brewer is a more accurate instrument in rejecting stray light effects (Gröbner et al., 1998) and as the single Brewers in ozone measurement stations around the world are being replaced by the more advanced double Brewer a slight increase in the ozone amounts may be observed particularly in those months in which observations must necessarily be made at large solar zenith angles. This may lead to a false assumption that the total ozone column amounts are showing an increasing trend where this may not be the case.

The departure from linearity is more prominent for large airmass ($\mu$) values as is seen in Figures 3, 5 and 7. This non-linearity at large airmasses becomes more prominent in the high latitude regions such as the Arctic where the sun is at large solar zenith angles most of the year. It is also difficult to take reference Brewer instruments to the Arctic for calibration purposes. To provide an accurate onsite method to correct the ozone data for non-linearity, the process described in this paper is recommended.

The future steps for this research will be to apply this method to historical data bases and produce an applicable software that would output corrected daily ozone values. Data processing using the new methodology leads to a more accurate absolute calibration on single Brewers and to a way to properly transfer calibration constants between single and double Brewers.

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

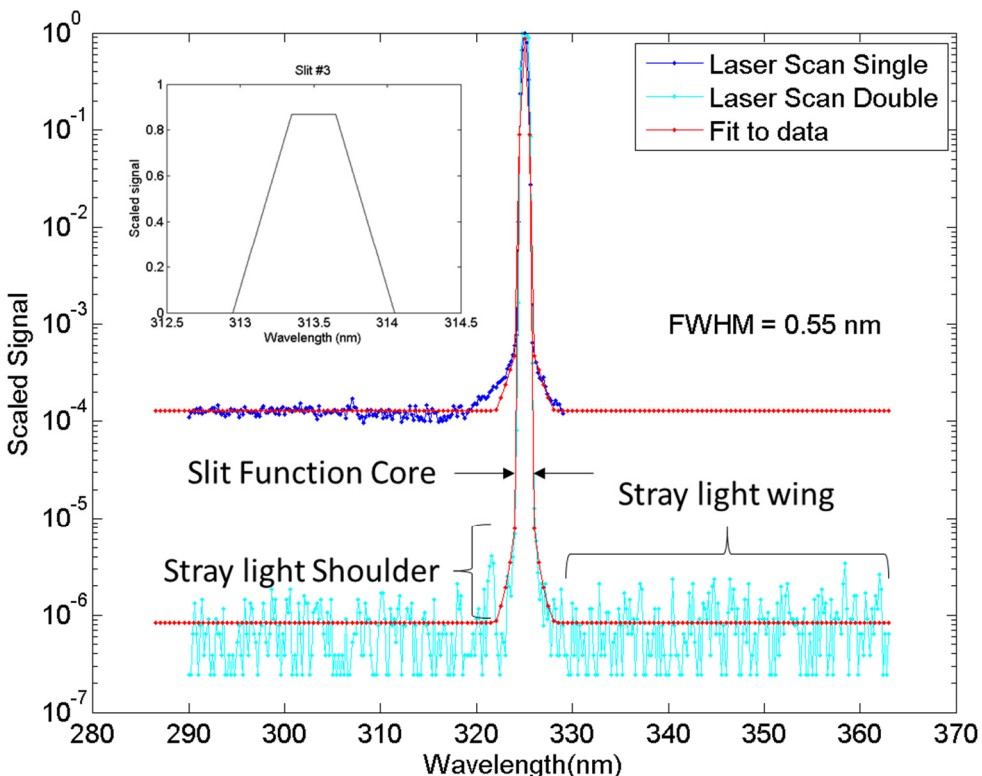

**Figure 1. Slit function measurements made with a HeCd-laser for single Brewer No. 009 and double Brewer No. 119 and their fitted slit functions. The ideal slit function is shown within the graph titled Slit#3. (Moeini, 2017)**

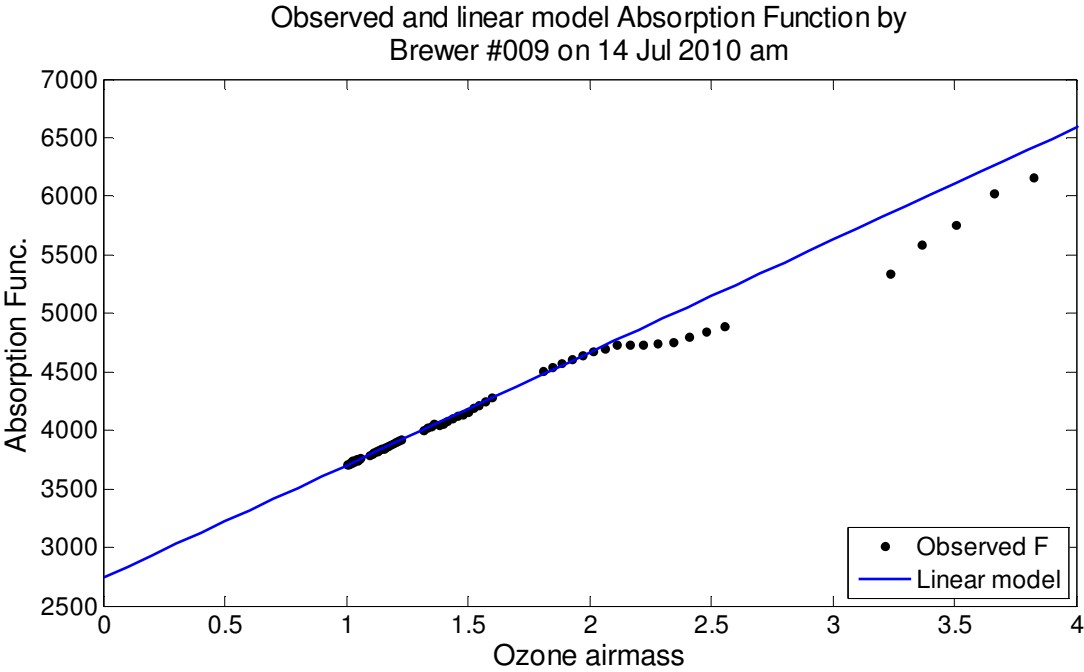

**Figure 2. Absorption function versus ozone airmass for the single Brewer #009. The fitted linear model is presented as the blue line and the black dots represent the observations.**

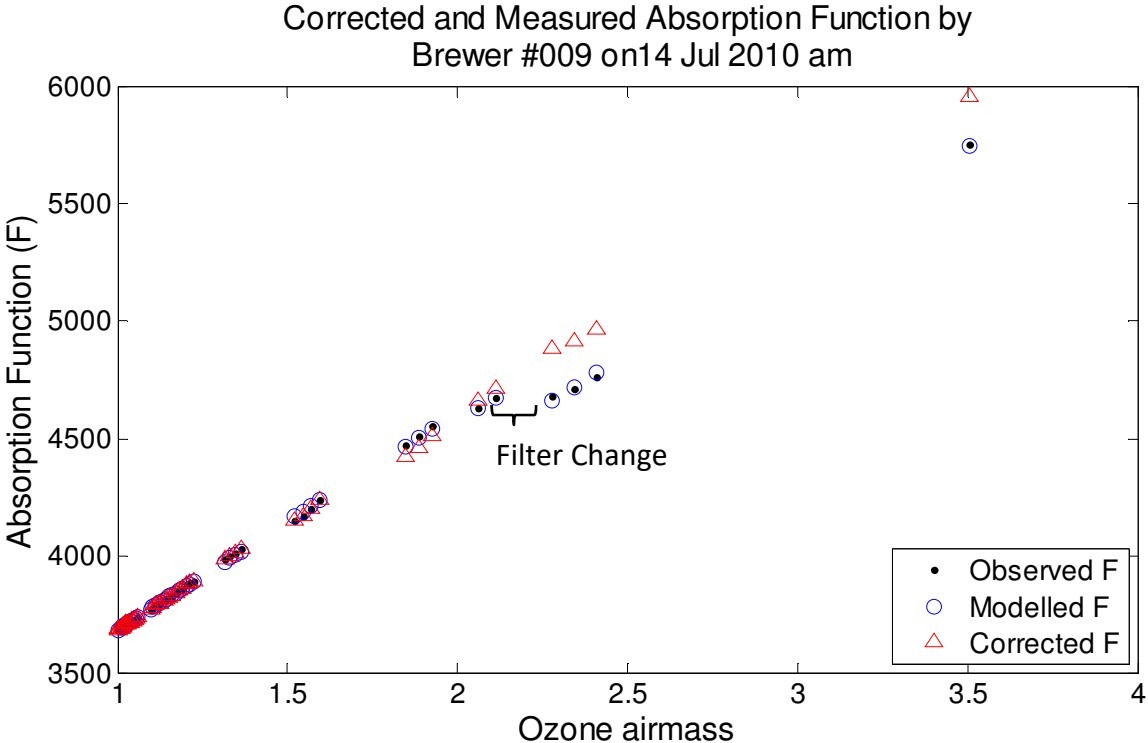

**Figure 3: Observed, Modelled and Corrected absorption function (F) for Single Brewer #009. The step change in observed F shows the filter change and the non-linear model corrects for the sudden drop in F.**

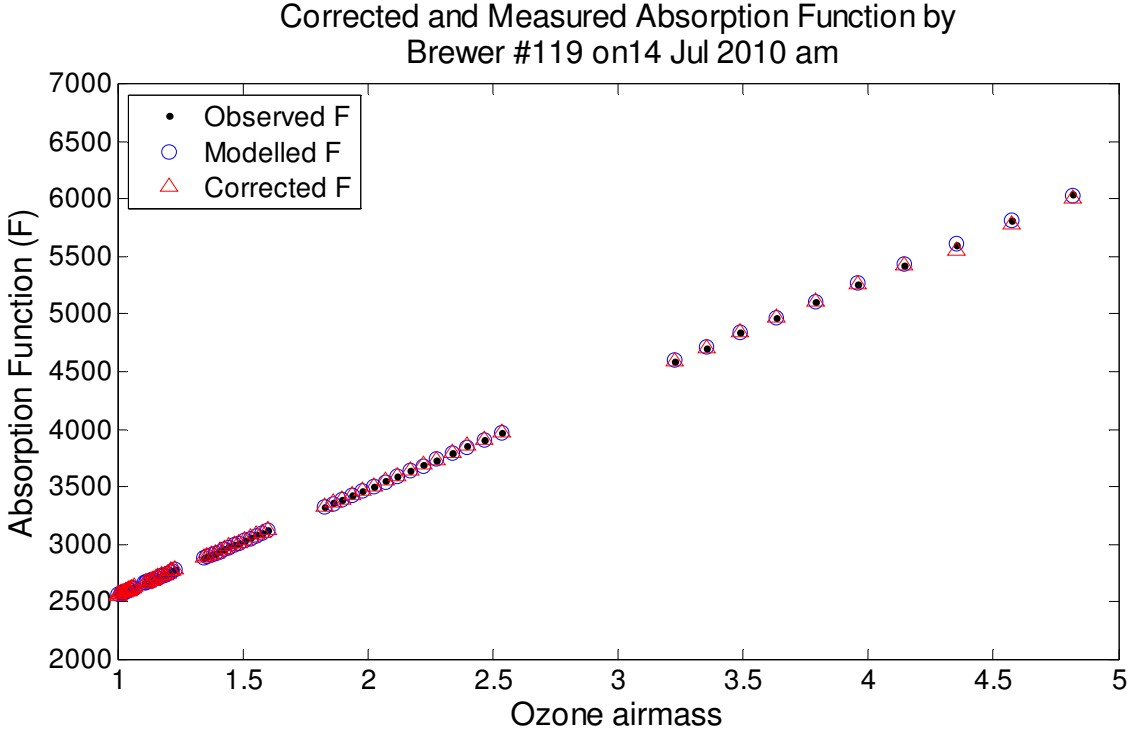

**Figure 4: Observed, Modelled and Corrected absorption function (F) for Double Brewer #119. The corrected values are very close to observed due to the Double Brewer's ability to reject stray light better than the Single Brewer.**

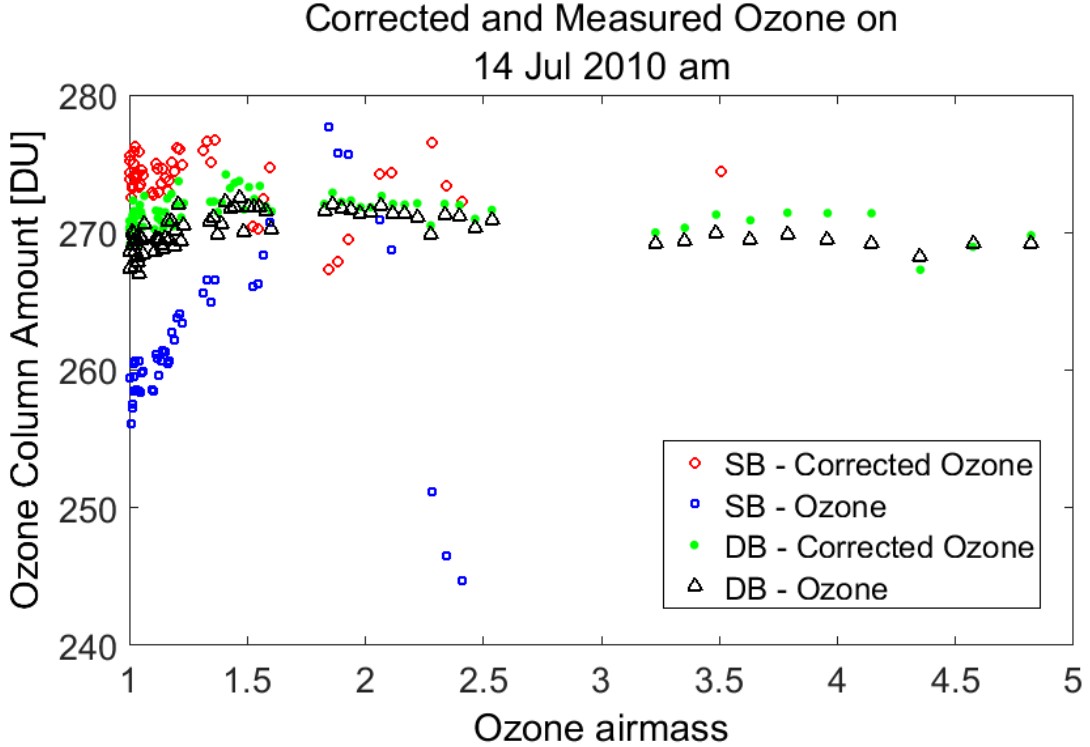

**Figure 5: Corrected and Measured Ozone by Single (SB #009) and Double (DB #119) Brewer on 14 July 2010.**

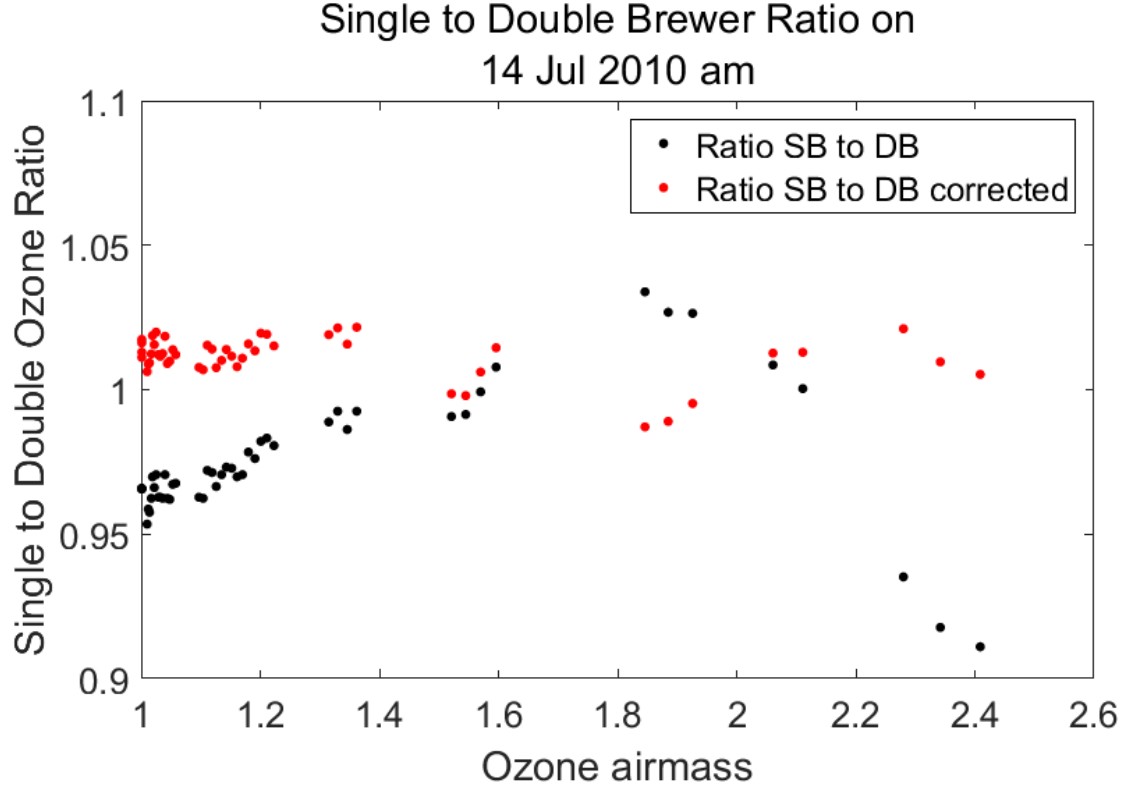

5    **Figure 6: Ratio of measured and corrected Single (SB) to Double Brewer (DB) Ozone values for data points measured within 5 minutes of each other on 14 July 2010.**

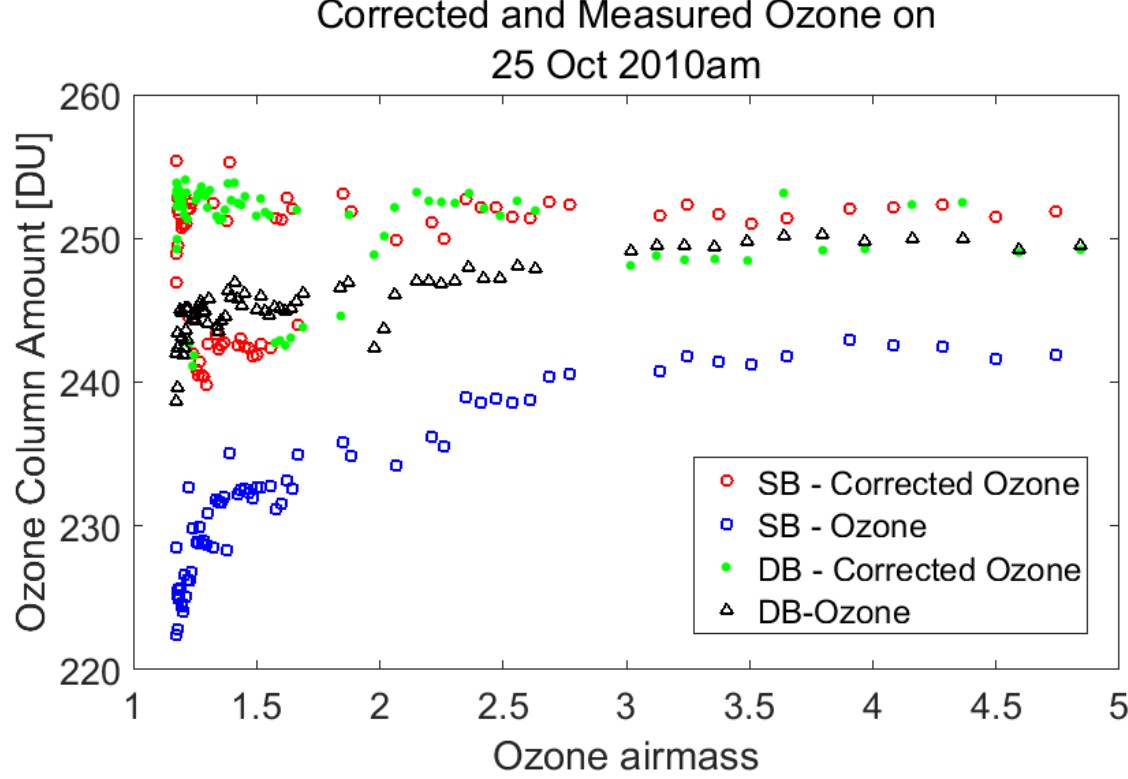

**Figure 7: Corrected and Measured Ozone by Single (SB #009) and Double (DB #119) Brewer on 25 Oct 2010.**

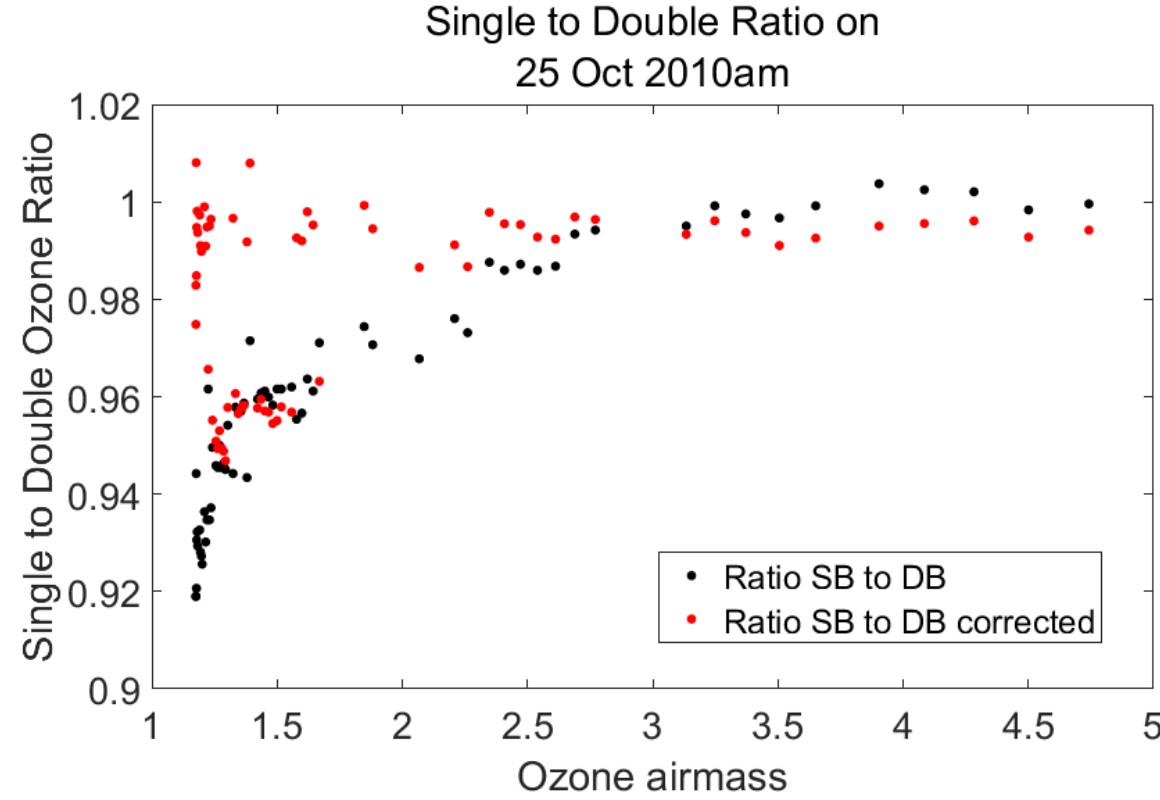

**Figure 8: Ratio of measured and corrected Single (SB) to Double Brewer (DB) Ozone values for data points measured within 5 minutes of each other on 25 Oct 2010.**

**Table 1: Standard ozone retrieval parameters compared to parameters retrieved from the model (14$^{th}$ July 2010).**

| Ozone retrieval parameters | Single #009 | | Double #119 | |
|---|---|---|---|---|
| | Standard | Model | Standard | Model |
| Extraterrestrial Constant (ETC) | 2855 | 2783.8 | 1647 | 1639.3 |
| Dead Time (DT) | $3.9\times10^{-8}$ | $3.27502\times10^{-9}$ | $4.2\times10^{-8}$ | $1.6\times10^{-9}$ |
| Non-linearity Factor (Ɣ) | - | $7.8\times10^{-17}$ | - | $2.7\times10^{-17}$ |

## 6. Appendix

Jacobian Calculations of $M_{ik}$:

$$M_{ik} = \frac{\partial(F_{mi})}{\partial v_k}$$

$$M_{i1} = \frac{\partial(F_{mi})}{\partial x} = -\alpha \cdot \mu - 3\gamma \cdot (\alpha \cdot \mu)^3 \cdot (x)^2$$

$$M_{i2} = \frac{\partial(F_{mi})}{\partial \gamma} = -(\alpha \cdot \mu \cdot x)^3$$

$$M_{i3} = \frac{\partial(F_{mi})}{\partial b_j} = ND_j$$

$$M_{i4} = \frac{\partial(F_{mi})}{\partial F_0} = 1$$

$$M_{i5} = \frac{\partial(F_{mi})}{\partial \tau} = -10^4 \cdot w \cdot C_\lambda$$