# Peer review of "A Calibration Procedure Which Accounts for Non-linearity in Singlemonochromator Brewer Ozone Spectrophotometer Measurements"

_Atmospheric Measurement Techniques, 2018_

## Referee Comment (RC1) · Anonymous Referee #2 · 24 Jul 2018

This paper deals with the very important topic of accounting for the errors introduced into stratospheric ozone measurements by the effects of stray light on the Brewer Ozone Spectrophotometer. The method described is novel and deals with the issue at source during the primary calibration by Langley Plot. The paper should be published with the following minor revisions:

There have been other studies into stray light effects which should be discussed in relation to this work and referenced. e.g. a co-author also contributed to:

[Figure]

Karppinen, T., Redondas, A., García, R. D., Lakkala, K., McElroy, C. T., and Kyrö, E.: Compensating for the Effects of Stray Light in Single-Monochromator Brewer Spectrophotometer Ozone Retrieval, Atmosphere-Ocean, 53, 66-73, 2015.

Also the new European Brewer Network (EuBrewNet) is already applying stray light correction in its data processing algorithms. See:

Rimmer, J. S., Redondas, A., and Karppinen, T.: EuBrewNet – A European Brewer network (COST Action ES1207), an overview, Atmos. Chem. Phys., 18, 10347-10353, 2018.

Redondas, A., Carreño, V., León-Luis, S. F., Hernández-Cruz, B., López-Solano, J., Rodriguez-Franco, J. J., Vilaplana, J. M., Gröbner, J., Rimmer, J., Bais, A. F., Savastiouk, V., Moreta, J. R., Boulkelia, L., Jepsen, N., Wilson, K. M., Shirotov, V., and Karppinen, T.: EUBREWNET RBCC-E Huelva 2015 Ozone Brewer Intercomparison, Atmos. Chem. Phys., 18, 9441-9455, 2018.

Specific points:

P1 L16 An ozone column of 600DU would not be typical??

P1 L19 Primary calibrations are also performed at the Izana Observatory in Tenerife.

P2 L19 The ref 'Bais et al. 1996' refers to spectral UV measurements where measurements < 300nm suffer from stray light effects. Brewer ozone measurements are taken at fixed wavelengths longer than 300nm, as stated in the introduction, where you say the difference is insignificant. Some clarification is required here.

P3 L28 I think the paper would benefit from a better description of the application of weighting coefficients, or at least a reference such as:

Savastiouk, V. and McElroy, C. T.: Brewer spectrophotometer total ozone measurements made during the 1998 Middle Atmosphere Nitrogen Trend Assessment (MANTRA) campaign, Atmosphere-Ocean, 43, 315-324, 2005.

P6 L1 Effects of deadtime could be referenced e.g.:

Fountoulakis, I., Redondas, A., Bais, A. F., Rodriguez-Franco, J. J., Fragkos, K., and Cede, A.: Dead time effect on the Brewer measurements: correction and estimated uncertainties, Atmos. Meas. Tech., 9, 1799-1816, 2016.

Technical:

P4 L17 There are no red dots in figure 2.

P4 L19 Shouldn't the slope of the Langley be +alpha.x ?

P5 L3 Delta Fi is substituted in eqn 2.7 not 2.8

Finally, there does not seem to be a reference to figure 6 in the text.

---

## Referee Comment (RC2) · Anonymous Referee #1 · 2 Aug 2018

This is the first review of the manuscript "A Calibration Procedure Which Accounts for Non-linearity in Single monochromator Brewer Ozone Spectrophotometer Measurements" by Zahra Vaziri Zanjani et al. This paper addresses a very important subject of the stray light interference in observations. Authors describe methods to determine stray light contribution in the Brewer total ozone column measurement under low sun conditions. The paper provides a very detail mathematical description of the Brewer measurement principles and the method to evaluate instrumental parameters that contribute to the non-linearity of total ozone derivation. The paper lacks in discussion of

the results. It provides only one case study (one day) to show that the approach works. If the mathematical model is correct, the fit to the single day of observations should work very well. Thus, the good fit is not a surprise. It is more important to demonstrate that the mathematical model can correct stray light continuously over the extended period of observations. It would benefit the reader to learn how well the mathematical model (determined from the limited set of data) would work if applied to a series of observations.

Several technical suggestion and comments 1) DB – may be change to DBr to avoid mixing with Dobson abbreviations. Then SB also should change to SBr for similarity 2) P.3 line 7 "The difference between the wings of the two slit functions" – did you mean it is a difference between measured and idealized? Please clarify. 3) P. 3 lines 18-19 "In the Brewer instrument, I0 is ..." it is true not just for Brewer instrument, but also for any ground-based instrument that uses source of light to make radiative flux measurements.... May be change to the "The absolute intensity measurement by the ground-based instrument relies on the knowledge of the extra-terrestrial source of the light", or similar 4) P. 3, line 21-23 It is called differential absorption methods, also used in Dobson instrument. 5) P. 4 Eq. 2.1-2.4 – transition to eq. 2.4 is not clear (from intensities measured at two wavelength to the counts). 6) P. 4, equation 2. 5. Is absorption function written for one spectral channel of the Brewer instrument? Should alfa be dependent on the wavelength? Should there be a sum of the bj*NDj for all filters? 7) P. 4 line 11 – what does k represent? Is it number of Brewer spectral channels or number of observations? 8) P. 6, eq. 2.16 – This equation is compared to the Eq. 2.8. What is the order of this correction? Is it applied prior to iteration process (Eq. 2.8-2.13)? 9) P. 6. "Illustrated in Figure 5". Please add discussion of results shown in Figure 5. 10) Figure 6 is not discussed in the text.

---

## Author Comment (AC1) · 4 Sep 2018

1) Pg 1, Ln 26-32, references and a short discussion of the papers mentioned has been added. 2) It is just and example of how a very large slant column amount can occur. Ozone columns in excess of 500 DU are not unusual in the Arctic spring when measurements still must be made at large solar zenith angles. Clarification has been made in the text, Pg 1, Ln 18 3) Izana Observatory has been added, Pg 1 Ln 19 4) At wavelengths above 300 nm, the presence of stray light may be problematic as well. This paper addresses this issue. Pg 2, Ln 29 5) A reference to the weighting

coefficients has been added. Going into more detail about this is outside the scope of this paper. Pg 4, Ln 14 6) Effects of deadtime have been referenced. Pg 6, Ln 15 7) Corrected the dots. Pg 5, Ln 7 8) The slope of Langley plot is alpha.x which has been corrected. Pg 5, Ln 9 9) Eq 2.8 has been corrected. Pg 5, Ln 19 10) Reference to figure 6 has been added. Pg 7, Ln 17-23

Please also note the supplement to this comment:
https://www.atmos-meas-tech-discuss.net/amt-2018-157/amt-2018-157-AC1-supplement.pdf

---

## Author Comment (AC2) · 4 Sep 2018

An additional day of data has been included in the paper to show the mathematical model works with more than one day of data. The application of this method to a series of data is part of the future work intended and is beyond the intended scope of this paper. 1) DB and SB have been changed to DBr and SBr. 2) Pg 3, Ln 17, the difference between the wings has been clarified. 3) Pg 3, Ln 30, I0 has been corrected. 4) Pg 3, Ln 33, differential absorption method added. 5) There is no transition. F is calculated from the counts and substituted in Eq 2.5 6) One alpha is calculated for

each instrument, the sum of the filters is used and is now shown in the model equation. 7) Pg 5, Ln 1, k is the number of components of v that are to be retrieved added to the text. 8) This correction is applied at the same time as the others. 9) Discussion of results has been clarified and figure 6 has been called out.

Please also note the supplement to this comment:
https://www.atmos-meas-tech-discuss.net/amt-2018-157/amt-2018-157-AC2-supplement.pdf

**Supplement:**

[revised manuscript text omitted]

---

## Author Response (AR1)

**Reply to Referee #1**

An additional day of data has been included in the paper to show the mathematical model works with more than one day of data. The application of this method to a series of data is part of the future work intended and is beyond the intended scope of this paper.

1) DB and SB have been changed to DBr and SBr.
2) Pg 3, Ln 17, the difference between the wings has been clarified.
3) Pg 3, Ln 30, $I_0$ has been corrected.
4) Pg 3, Ln 33, differential absorption method added.
5) There is no transition. F is calculated from the counts and substituted in Eq 2.5
6) One alpha is calculated for each instrument, the sum of the filters is used and is now shown in the model equation.
7) Pg 5, Ln 1, k is the number of components of $v$ that are to be retrieved added to the text.
8) This correction is applied at the same time as the others.
9) Discussion of results has been clarified and figure 6 has been called out.

**Reply to Referee #2**

1) Pg 1, Ln 26-32, references and a short discussion of the papers mentioned has been added.
2) It is just and example of how a very large slant column amount can occur.  Ozone columns in excess of 500 DU are not unusual in the Arctic spring when measurements still must be made at large solar zenith angles.  Clarification has been made in the text, Pg 1, Ln 18
3) Izana Observatory has been added, Pg 1 Ln 19
4) At wavelengths above 300 nm, the presence of stray light may be problematic as well. This paper addresses this issue. Pg 2, Ln 29
5) A reference to the weighting coefficients has been added. Going into more detail about this is outside the scope of this paper. Pg 4, Ln 14
6) Effects of deadtime have been referenced. Pg 6, Ln 15
7) Corrected the dots. Pg 5, Ln 7
8) The slope of Langley plot is alpha.x which has been corrected. Pg 5, Ln 9
9) Eq 2.8 has been corrected. Pg 5, Ln 19
10) Reference to figure 6 has been added. Pg 7, Ln 17-23

[revised manuscript text omitted]
*, 53(1), pp.66–73. Available at: http://dx.doi.org/10.1080/07055900.2013.871499, 2015.

Kerr, J.B., McElroy, C.T., Wardle, D.I., Olafson, R.A., Evans, W.F.J.: The automated Brewer spectrophotometer. In C. S. Zerefos & A. Ghazi, eds. *Quadrennial Ozone Symposium*. Halkidiki, Greece, pp. 396–401, 1984

Kerr, J.B.: The Brewer Spectrophotometer. In W. Gao, D. L. Schmoldt, & J. R. Slusser, eds. *UV Radiation in*

*Global Climate Change*. Tsinghua University Press, Beijing and Springer, pp. 160–191, 2010

Moeini, O., Vaziri, Z., McElroy, C.T., Tarasick, D.W., Evans, R.D., Petropavlovskikh, I., Feng, K.H.: The Effect of Instrumental Stray Light on Brewer and Dobson Total Ozone Measurements. Submitted to: *Atmospheric Measurement Techniques*, 2018.

5    Redondas, A., Carreno, V., Leon-Luis, S.F., Hernandez-Cruz, B., Lopez-Solano, J., Rodriguez-Franco, J.J., Vilaplana, J.M., Grobner, J., Rimmer, J., Bais, A.F., Savastiouk, V., Moreta, J.R., Boulkelia, L., Wilson, K.M., Shirotov, V., Karppinen, T.: EUBREWNET RBCC-E Huelva 2015 Ozone Brewer Intercomparison. *Atmospheric Chemistry and Physics*, 18(13), pp.9441–9445, 2018.

Redondas, A., Evans, R., Stuebi, R., Kohler, U., Weber, M.: Evaluation of the use of five laboratory-
10    determined ozone absorption cross sections in Brewer and Dobson retrieval algorithms. *Atmospheric Chemistry and Physics*, 14(3), pp.1635–1648, 2014.

Rimmer, J.S., Redondas, A., Karppinen, T.: EuBrewNet - A European Brewer network (COST Action ES1207), an overview. *Atmospheric Chemistry and Physics*, 18(14), pp.10347–10353, 2018.

Savastiouk, V.: *Improvements To the Direct-Sun Ozone Observations Taken With the Brewer*
15    *Spectrophotometer*. York University. Available at: https://www.esrl.noaa.gov/gmd/grad/neubrew/docs/publications/VladimirSavastiouk_PhD_thesis.pdf., 2016.

Savastiouk, V., McElroy, C.T.: Brewer spectrophotometer total ozone measurements made during the 1998 Middle Atmosphere Nitrogen Trend Assessment (MANTRA) Campaign. *Atmosphere - Ocean*, 43(4),
20    pp.315–324, 2005.

Silva, A.A., Kirchhoff, V.W.J.H.: Aerosol optical thickness from Brewer spectrophotometers and an investigation into the stray-light effect. *Applied Optics*, 43(12), pp.2484–2489, 2004.

Slavin, W.: Stray Light in Ultraviolet, Visible and Near-Infrared Spectrophotometry. *Analytical Chemistry*, 35(4), pp.561–566, 1963.

25    WOUDC: No Title. *WOUDC*. Available at: http://www.woudc.org/data/instruments/ [Accessed May 1, 2017], 2016.

[Figure]

**Figure 1. Slit function measurements made with a HeCd-laser for single Brewer No. 009 and double Brewer No. 119 and their fitted slit functions. The ideal slit function is shown within the graph titled Slit#3. (Moeini, 2017)**

[Figure]

**Figure 2. Absorption function versus ozone airmass for the single Brewer #009. The fitted linear model is presented as the blue line and the black dots represent the observations.**

[Figure]

**Figure 3: Observed, Modelled and Corrected absorption function (F) for Single Brewer #009. The step change in observed F shows the filter change and the non-linear model corrects for the sudden drop in F.**

[Figure]

**Figure 4: Observed, Modelled and Corrected absorption function (F) for Double Brewer #119. The corrected values are very close to observed due to the Double Brewer's ability to reject stray light better than the Single Brewer.**

[Figure]

**Figure 5: Corrected and Measured Ozone by Single (SB #009) and Double (DB #119) Brewer on 14 July 2010.**

[Figure]

5   **Figure 6: Ratio of measured and corrected Single (SB) to Double Brewer (DB) Ozone values for data points measured within 5 minutes of each other on 14 July 2010.**

[Figure]

Figure 7: Corrected and Measured Ozone by Single (SB #009) and Double (DB #119) Brewer on 25 Oct 2010.

[Figure]

Figure 8: Ratio of measured and corrected Single (SB) to Double Brewer (DB) Ozone values for data points measured within
5 minutes of each other on 25 Oct 2010.

---

## Referee Report (RR1)

Alberto Redondas:

Referee Report A Calibration Procedure Which Accounts for Non-linearity in Single-monochromator Brewer Ozone Spectrophotometer Measurements Zahra Vaziri Zanjani, Omid Moeini, Tom McElroy, David Barton, and Vladimir Savastiouk

This paper deals with an important topic of accounting for the errors introduced into stratospheric ozone measurements by the effects of stray light on the Brewer Ozone Spectrophotometer. The method allows obtaining not only stray light characterization but also instrumental parameters like the Dead Time using the Langley Plot method of calibration. The method is novel and good presented but not enough tested, is only discussed with two days of observations.

 The paper should be published with the following minor revision.

1. The absorption function calculation is not enough described, in particular how are the intensities are calculated from the PMT counts. I think is important as counts, dead time and neutral density filters are used on the model. Can this method can be applied to the determination of the temperature coefficients ?.
2. A good indication of the model could be the comparison of the obtained parameters with ones used on the standard ozone retrieval.  In particular is the differences in the Extraterrestrial constant (Fo).
3. I also miss the discussion on how the different parameters affect ozone, splitting the influence of straylight, neutral density filters or dead time.   This will help the discussion obtained results. For example, also the modeled double brewer increase the ozone at noon, this is due DT or is a filter effect ?.

4. I suggest showing as an appendix the Jacobian calculations (Mik), who can illustrate the influence of the different parameters.
5. To get a Langley the ozone, and any other interfering species should be constant during the measurement period, how this effect if this condition is not meet, as the observations on 25 of October were the ozone change about ~10 DU  on the double brewer.

Oher technical comments:

Page 1 Line 25: References needed for the mentioned studies.

Page 1 Line 32: "Both" are confusing,  Karppinen method is not based on double brewer comparison.

Page 2 Line 5:  It requires langley conditions (stable)  so this limits the applicability of the method.

Page 2 Line 20:  There is another source of straylight, the atmospheric due field of view of the instrument (see for example Jossefson 1992).

Page 2 Line 30;  Bais is not dealing with ozone on his paper, the 10% error addressed must be referenced

Page 3 Line 10 % Brewer wavelength changing from instrument to instrument.

Page 4, Line 27: ND filter vector is not defined

Page 6, Line 1 : There is no description of the convergence,  number of iterations needed or the condition to finish the the interation.

Page 6  Line 5: The photon counts on the spectrometer are related with the neutral density filter used, is not unusual that the counts are lower during the high sun, due the use of high attenuation filter. A plot with the weighted/unweighted observations could clarify this. Dobson Langley use the regression against 1/m to avoid this effect (Kiedrom & Michalsky 2016)

Page 7 Line 6: The DT procedure is usually done daily  (Granjar et al  2008)

Alberto Redondas

References

Grajnar, T., Savastiouk, V., and McElroy, T.: Brewer Standard Operative Procedure (Draft) http://www.io3.ca/Download/Brewer_SOP_DRAFT.pdf (last access: 20 October 2015), 2008.

T. Pulli, T. Karppinen, S. Nevas, P. Kärhä, K. Lakkala, J. M. Karhu, M. Sildoja, A. Vaskuri, M. Shpak, F. Manoocheri, L. Doppler, S. Gross, J. Mes & E. Ikonen. Out-of-Range Stray Light Characterization of Single-Monochromator Brewer Spectrophotometers

Josefsson, W.: Focused sun observations using a Brewer ozone spectrophotometer, 97, 15 813–15817,https://doi.org/10.1029/92JD01030,https://agupubs.onlinelibrary.wiley.com/doi/abs/10.1029/92JD01030.5

KIEDRON, P. W.; MICHALSKY, J. J. Non-parametric, and least squares Langley plot methods. Atmospheric Measurement Techniques, 2016, vol. 9, no 1, p. 215-225.

---

## Author Response (AR2)

**Revisions:**

1) Page 4 Ln 15-20 explains how the counts from the photomultiplier are converted to absorption function.
2) Page 7 Ln 29-31 and table 1 have been added
3) The model combines all the different parameters which can affect the stray light and ozone measurements. Each parameter is separately retrieved but combined when calculating the corrected ozone.
4) Appendix added.
5) We only use the linear Langley condition (stable ozone) to predict an a priori value for the ETC value. The non-linear model retrieves the ETC value from fitting the non-linear model to the data.

**Technical comments:**

Page 1 Ln 25: Reference added.

Page 1 Ln 32: Clarified.

Page 2 Ln 5: We only use the linear Langley condition (stable ozone) to predict an a priori value for the ETC value. The non-linear model retrieves the ETC value from fitting the non-linear model to the data.

Page 2 Ln 20: sky-scattered radiation plus reference has been added.

Page 2 Ln 30: ozone changed to absolute irradiances, but this error also applies to ozone measurements which are derived from absolute irradiance.

Page 3 Ln 10: While each Brewer will have slightly different exact wavelengths, they are always very close together and the stray light effect will be very similar, and the impact will be absorbed into the non-linear correction parameter.

Page 4 Ln 27: ND has been defined

Page 6 Ln 1: page 5 Ln 20 has been added to define maximum number of iterations.

Page 6 Ln 5: We also multiply by 1/airmass. I corrected the typo.

Page 7 Ln 6: While daily DT measurements are frequently made, the changes in the result over a measurement period may not be reflected in the data analysis. In order to ensure an accurate calibration, the DT is retrieved as part of the calibration procedure.

[revised manuscript text omitted]